# Enhancing Multilingual Reasoning in LLMs: Insights from Cross-Linguistic Correlations and Optimal Data Proportions

**Jiangkuo Wang,**[*] **Suyv Ma,**[*] **Mingpeng Wei** [†]
DeepShare Inc.

## Abstract

Large language models (LLMs) typically rely on fine-tuning to enhance their reasoning capabilities across various languages. However, limited research has been conducted on the optimal balance of language proportions within multilingual reasoning datasets. To fill this gap, we performed a systematic study to examine how different proportions of language data in multilingual reasoning datasets influence fine-tuning performance. Our study revealed a clear relationship between language proportions in datasets and the fine-tuning performance of LLMs. By fine-tuning multiple LLMs using the appropriate language distributions and data volumes identified in our study, we achieved state-of-the-art performance in both multilingual mathematical reasoning and solving mathematical problems using Python code. Furthermore, our approach significantly reduced data volume requirements and translation costs compared to existing methods, providing a valuable reference for future research.

## 1 Introduction

Despite recent significant advancements in LLMs (OpenAI, 2023; Chowdhery et al., 2023; Touvron et al., 2023; Brown et al., 2020; Workshop et al., 2023), LLMs still encounter considerable challenges in reasoning tasks for non-English languages, particularly low-resource languages (Shi et al., 2022b; Huang et al., 2023b; Qin et al., 2023b). A common scenario is that during the post-training process of LLMs, it is necessary to simultaneously enhance their reasoning capabilities across multiple widely used languages. A critical research question thus emerges: how can LLMs, predominantly trained on high-resource languages like English, effectively generalize their reasoning capabilities to low-resource languages that suffer from insufficient training data? Addressing this question is essential for developing more inclusive and globally applicable AI systems.

Fine-tuning has been the commonly adopted solution to this issue. Prior studies have primarily focused on fine-tuning models using translated mathematical reasoning datasets across multiple languages. For instance, one approach enhanced LLMs' multilingual reasoning capabilities using translated datasets (Chen et al., 2023), while another improved model performance through question alignment (Zhu et al., 2024b;a). While these methods have improved multilingual reasoning performance, they still exhibit significant limitations. First, previous work often involved translating English datasets into multiple languages in equal proportions for fine-tuning. This approach is costly, overlooks the impact of language proportions in the fine-tuning datasets. Second, in scenarios requiring extensive multilingual data, it is impractical to translate English data into multiple languages in equal amounts. Thus, the key is to efficiently leverage a small amount of low-resource language data to broadly enhance the multilingual reasoning capabilities of LLMs.

Moreover, earlier studies mainly focused on a limited number of languages and rarely tested generalization capabilities across more than 20 languages. Additionally, most prior research was conducted on only a few LLMs, and it remains unclear whether these approaches generalize well to other, more advanced models.

---

[*]Work done while working as an intern at DeepShare Inc.

[†]Corresponding Author, Email: `Samuel@deepshare.ai`
 Code repository: `https://github.com/DeepShareAI/HighMath`

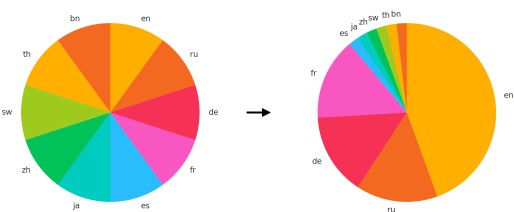

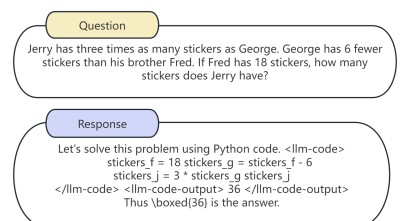

Figure 1: Two charts represent the proportion of data for different languages in the multilingual mathematical reasoning dataset. In previous methods, the amount of data for each language was the same (as shown in the left chart). Through our research, we determined the optimal proportion for each language (as shown in the right chart).

Figure 2: We use this data format to teach model to solve mathematical reasoning task with executable code.

In this paper, we address these challenges through a systematic investigation of how varying language proportions in multilingual mathematical reasoning datasets affect fine-tuning outcomes, and determine the optimal amount of data needed for fine-tuning, as illustrated in Figure 1.Directly exploring the optimal ratio among tens of languages is extremely challenging, as it requires determining the optimal proportions among tens of variables, which results in an extremely large search space. So we adopted an innovative approach to effectively reduce the search space and determined the appropriate ratio for the 10 languages commonly used in previous works (Zhu et al., 2024b;a), which was further extended to 25 languages. Additionally, we explored the impact of data volume on the fine-tuning performance of LLMs. Ultimately, building on previous research, we created the largest multilingual mathematical reasoning dataset, HighMath-350k, alongside a multilingual code reasoning dataset(Figure 2), HighCode-350k. Fine-tuning multiple LLMs on these datasets resulted in state-of-the-art performance, clearly demonstrating the efficacy of our approach.

In summary, our key contributions are as follows:

- Through more than 600 groups of experiments, we analyzed and modeled how the proportions of different languages in multilingual reasoning datasets affect fine-tuning performance, and determined the appropriate volume of fine-tuning data.

- We constructed multilingual chain-of-thought mathematical reasoning dataset, HighMath-350k, as well as the multilingual code reasoning dataset, HighCode-350k.

- Based on multiple LLMs, we trained state-of-the-art multilingual chain-of-thought reasoning models and code reasoning models.

## 2 RELATED WORK

### 2.1 MULTILINGUAL MATHEMATICAL REASONING

Mathematical reasoning is a core task for assessing the intelligence of LLMs (Zhang et al., 2024b;a), requiring them to understand mathematical problems and generate answers through step-by-step reasoning (Ahn et al., 2024; Zhang et al., 2022; Liu et al., 2023). (Shi et al., 2022b) expanded this evaluation to a multilingual setting by translating English math questions from the GSM8K test set (Cobbe et al., 2021) into various non-English languages, introducing the multilingual benchmark MGSM. Efforts to improve LLMs' multilingual reasoning performance have been continuing.For example, (Huang et al., 2023a) and (Qin et al., 2023a) explored prompting ChatGPT (OpenAI, 2023) to translate non-English questions into English and generate answers based on the translations. However, (Hu et al., 2024) found that this prompting strategy is not consistently effective for open-source LLMs. To enhance the multilingual capabilities of these models, researchers like (Nguyen et al., 2024) have explored continued pretraining on large-scale non-English corpora. However,

this approach is resource-intensive and data-inefficient, highlighting the need for more optimized fine-tuning strategies—a gap our study aims to address.

## 2.2 LLM's Language Preference

LLMs, with their large-scale parameters (Aki, 1967; Huo & Kassab, 2009; Wang et al., 2006; Rosenfeld, 1999), pre-trained on vast corpora and fine-tuned on comprehensive instruction datasets (Huang et al., 2016; Zhao et al., 2020; Lindström & Abraham, 2022; Koncel-Kedziorski et al., 2016), have demonstrated impressive intelligence (Touvron et al., 2023; Floridi & Chiriatti, 2020; Kalyan, 2023; Lagler et al., 2013). However, empirical studies show that LLMs still struggle with multilingual scenarios, particularly for low-resource languages (Shi et al., 2022a; Zhu et al., 2024b; Weyssow et al., 2024). This issue is mainly due to the dominance of English in both pretraining (Blevins & Zettlemoyer, 2022) and instruction datasets (Wang et al., 2023). In this work, we focus on a core capability of LLMs—their reasoning ability—and aim to push the boundaries of their performance in multilingual reasoning tasks.

## 3 METHODOLOGY

This section describes the systematic method used to determine the appropriate language proportions in the dataset and the required data volume for effective fine-tuning. Our method is divided into three phases, as detailed below, with the experimental setup provided in Section 4.1.

## 3.1 PHASE ONE: LANGUAGE ALIGNMENT ANALYSIS

Directly determining the optimal ratio among 10 languages is also unrealistic, as directly exploring the optimal proportions among 10 variables would result in a huge search space. To address this issue and better generalize the reasoning capabilities of LLMs to different language contexts, we first examined the correlation between LLMs' alignment with English and other languages, and their mathematical reasoning capabilities.Through this experiment, we aim to reduce the search space to an acceptable size.

This alignment was assessed through a Non-English to English translation task. First, several LLMs were fine-tuned on the English mathematical reasoning dataset, MetaMathQA-395k (Yu et al., 2024), to evaluate their translation capabilities from Non-English to English, both before and after fine-tuning. These models were then fine-tuned on translation data from non-English to English, and their performance on multilingual mathematical reasoning tasks was evaluated.Our results showed a positive correlation between LLMs' reasoning abilities and their alignment between English and French, German, and Russian, while no significant correlation was found with other languages.This finding will greatly help the subsequent experiments.

## 3.2 PHASE TWO:LANGUAGE GROUP OPTIMIZATION

Building on the insights from Phase One, we constructed a multilingual mathematical reasoning dataset encompassing 10 languages and grouped these 10 languages into three groups(as shown in Table 3) based on the findings from Phase One, treating them as three variables for further exploration. By systematically varying the proportions of data for each group, we utilized five models and conducted over 600 experiments to identify the optimal distribution of three groups. The findings indicate that, given a fixed total data volume, the optimal language ratio for enhancing the multilingual reasoning capabilities of LLMs is group1:group2:group3 = 4:4:1 (en:ru:de:fr:es:ja:zh:bn:sw = 24:8:8:8:1:1:1:1:1:1). Additionally, we employed Gaussian Process Regression (GPR) to model the relationship between data proportions and fine-tuning outcomes. This statistical approach allows us to predict optimal language distributions based on experimental data, thereby guiding efficient data allocation in subsequent phases. We also evaluated the contribution of each language to the overall improvement of multilingual reasoning.

### 3.3 PHASE THREE: DATA VOLUME DETERMINATION

To determine the appropriate volume of fine-tuning data, we first conducted experiments with 10 languages and later extended the study to 25 languages. We tested varying data volumes and compared the results with previous approaches that used equal data volumes across languages. The experiments included fine-tuning data volumes up to 9.875M and scaling model parameters to 70B, demonstrating the clear advantages of our proposed methodology.

## 4 EXPERIMENT

### 4.1 OPTIMAL PROPORTION AND DATA VOLUME SELECTION FOR MULTILINGUAL FINE-TUNING

This section describes the experimental setup and key results for the three phases outlined in the methodology.

#### 4.1.1 EXPERIMENTAL SETUP

**Base LLMs**

Different base LLMs were selected for each phase based on the research objectives. The specific models are listed in Table 1.

Table 1: Base LLMs used in different phases

| Phase | Base LLMs |
|---|---|
| Phase 1 | LLaMA3.1-8B, LLaMA3-8B, Phi3.5-mini-instruct, Mistral-7B-v0.3, Qwen2-7B |
| Phase 2 | LLaMA3.1-8B, LLaMA3-8B, Phi3.5-mini-instruct, Mistral-7B-v0.3, Qwen2-7B |
| Phase 3 | LLaMA3-8B, LLaMA3-70B |

**Experimental Languages**

The selection of test languages for each phase was guided by specific research objectives and computational constraints. In Phase One, the objective was to investigate the relationship between alignment with English and the mathematical reasoning capabilities of various non-English languages. To achieve this, We primarily selected several high-resource non-English languages. Phase Two employed the 10 languages from the MGSM dataset, while Phase Three aimed to assess the generalizability of our approach across 25 widely spoken languages. The languages chosen for each phase are detailed in Table 2.

Table 2: Experimental languages used in different phases

| Phase | Experimental Languages |
|---|---|
| Phase 1 | en, ru, de, fr, es, ja, zh, sw, th, bn, lt, cs, ka, ar |
| Phase 2 | en, ru, de, fr, es, ja, zh, sw, th, bn |
| Phase 3 | zh, en, es, hi, ar, pt, bn, ru, ja, pa, jv, de, ko, fr, te, vi, tr, ta, it, fa, ur, mr, sw, th, pl |

We translated both the questions and answers in the multilingual chain-of-thought mathematical reasoning dataset. To ensure that the generated multilingual data effectively enhanced the model's reasoning capabilities, we used the powerful open-source model DeepSeek-Chat-v2-236B (DeepSeek-AI, 2024) for high-quality translations, adjusting its hyperparameters for this task. To maintain translation quality and consistency, we implemented the following strategies:

- Mathematical formulas were preserved, and all numbers were converted to Arabic numerals to facilitate cross-linguistic prediction.
- To improve translation accuracy, we included two examples in the prompts for each language.

For the mathematical reasoning with code dataset, we only translated the question parts since the solution involves Python code.

To ensure consistency across languages, we extracted all mathematical expressions from the translated questions and answers. If the calculations were correct and matched the English version, the translation was deemed accurate. If errors persisted across five consecutive translations, the corresponding data was discarded.

This approach ensures coherence and accuracy in the translation process, enabling comprehensive evaluation and application of the dataset in a multilingual context, while maintaining both linguistic and mathematical integrity.

**Tasks and Datasets**

In terms of test task selection, in Phase One, we used a translation task to assess the alignment between English and non-English languages within the LLM. In the subsequent phases, chain-of-thought mathematical reasoning tasks were used to evaluate the LLM's reasoning abilities.

For the training datasets, Phase One used the MetaMathQA-395k and WMT datasets. In later phases, we translated the 395k entries from MetaMathQA into 25 languages, generating a total of 9.875M entries. In Phase Two, 200k entries from each language were selected, totaling 1M entries. Phase Three used the full 9.875M dataset.

For the test datasets, Phase One used the MGSM dataset to evaluate the LLMs' multilingual mathematical reasoning capabilities, and 3,000 entries from the WMT dataset were selected to assess translation performance. The WMT dataset is a collection of parallel texts designed to advance machine translation systems. In the subsequent phases, the MGSM dataset was used as the primary test dataset.

### 4.1.2 MAIN RESULTS

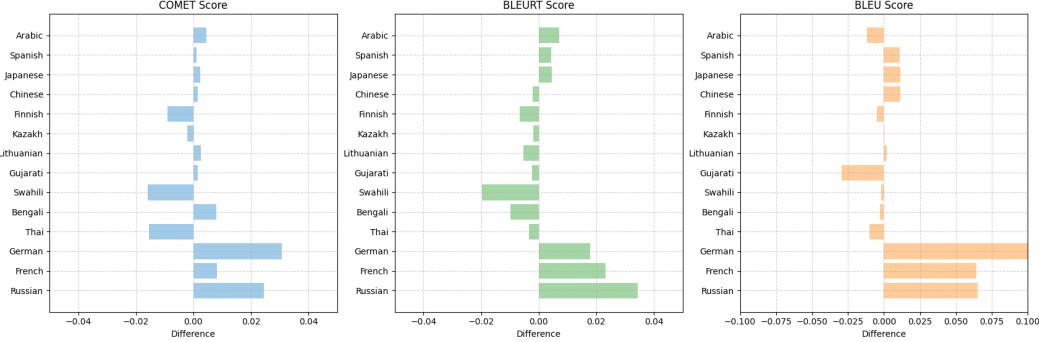

Figure 3

**Phase 1**

Figure 3 illustrates the changes in translation performance across different languages before and after fine-tuning the LLMs on the MetaMathQA-395k dataset. Significant improvements were observed for German, Russian, and French, while performance in other languages remained mostly unchanged.

Figure 4 presents the model's performance on the MGSM dataset after fine-tuning with different non-English to English translation pairs. Since no additional measures were taken to enhance reasoning ability, mathematical reasoning performance declined post-fine-tuning. However, German, Russian, and French still outperformed other languages in reasoning tasks.

These results suggest a positive correlation between the LLM's reasoning ability and its alignment with languages like French, German, and Russian.

**Phase 2**

Based on the conclusions from the first phase, we divided the 10 languages selected for the second phase into three groups, as shown in Table 3. In each group, the amount of mathematical reasoning

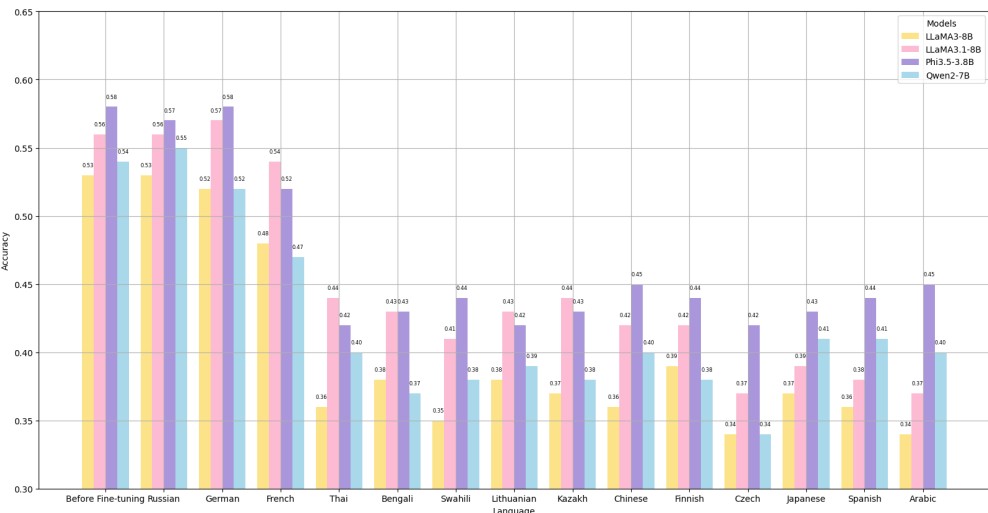

Figure 4: The score in the image represents the average of the LLM's scores across the 10 languages.

Table 3: Language Groups for the Experiment

| Group | Languages |
|-------|-----------|
| Group 1 | en |
| Group 2 | de, ru, fr |
| Group 3 | es, ja, zh, th, bn, sw |

data for each language was kept consistent. By adjusting the overall data ratio among these three groups, we explored the optimal distribution of languages. As shown in Figure 5, although the pretraining corpora for each model differ, all five models demonstrated optimal performance when the ratio of the three language groups was around 24:24:6.(4:4:1)

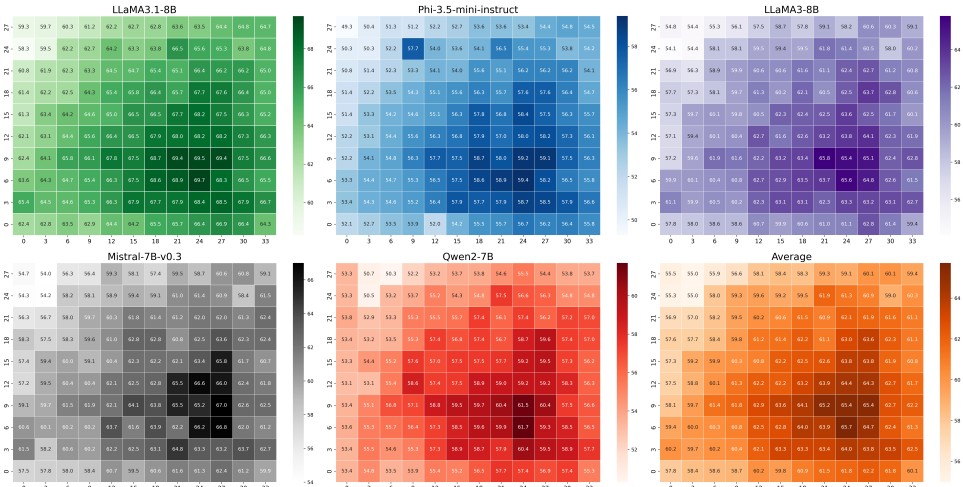

Figure 5: The relative proportion of English is fixed at 24. The horizontal axis represents the relative proportion of the second group of languages, while the vertical axis represents the relative proportion of the third group of languages. The results are averaged across the 10 languages.

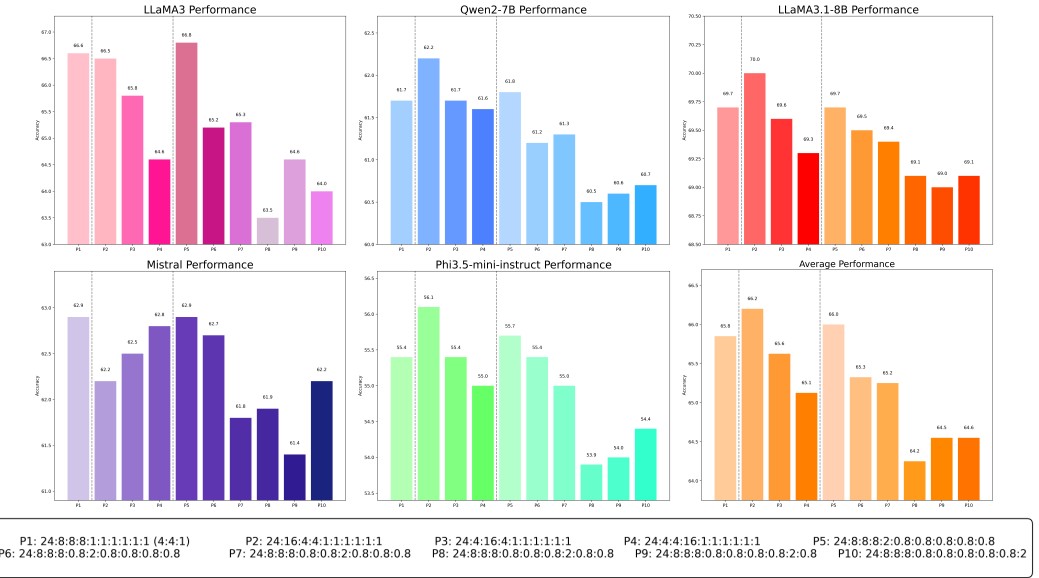

Figure 6: The proportions in the figure are represented as en:ru:de:fr:es:ja:zh:bn:sw:th in sequence.

**Phase 2**

To refine the optimal data ratio, we conducted further experiments to assess the relative importance of languages in the second and third groups. As shown in Figure 6, increasing the proportion of a single language did not lead to any clear improvement in prediction accuracy. Even when improvement occurred, the gains were minimal.

**Mathematical Modeling**

To better model the patterns observed in Figure 5 and provide suggestions for data allocation in future studies, we used the Gaussian Process Regression method (GPR). GPR is employed to model the underlying function $f(x)$ from the observed data, assuming that any finite set of function values follows a multivariate Gaussian distribution. The model can be expressed as:

$$\mathbf{y} = f(X) + \epsilon, \quad \epsilon \sim \mathcal{N}(0, \sigma^2 I),$$

where $X$ represents the input features, $\mathbf{y}$ is the observed output, and $\epsilon$ is Gaussian noise with variance $\sigma^2$.

We experimented with various mathematical models and computational methods, with detailed analyses provided in the appendix. Ultimately, we finalized the following kernel:

$$k_{\text{RQ}}(X, X') = \sigma_f^2 \left(1 + \frac{\|X - X'\|^2}{2\alpha l^2}\right)^{-\alpha}$$

**Phase 3**

This phase of the experiment was divided into two groups. The first group used the original 10 languages to construct different sizes of the multilingual mathematical reasoning dataset, High-Math, with a language ratio of en:ru:de:fr:es:ja:zh:bn:sw:th = 24:8:8:8:1:1:1:1:1:1. The second group included 25 languages and constructed different sizes of the HighMath dataset, using a ratio of en:de:ru:fr:others = 24:8:8:8:12.

We conducted a series of experiments, with the results shown in Figure 7. HighMath-350k and HighMath-1.5M was identified as a relatively optimal size.

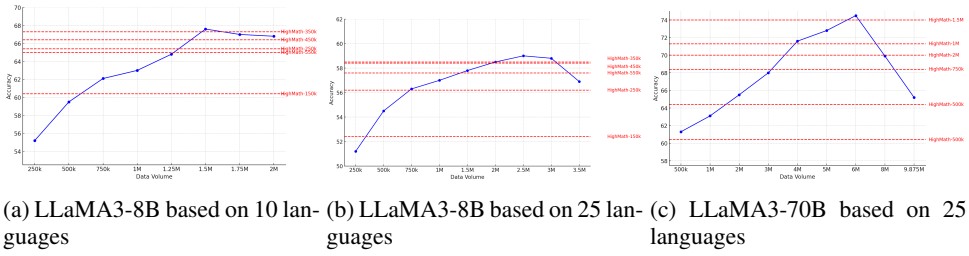

(a) LLaMA3-8B based on 10 languages  (b) LLaMA3-8B based on 25 languages  (c) LLaMA3-70B based on 25 languages

Figure 7

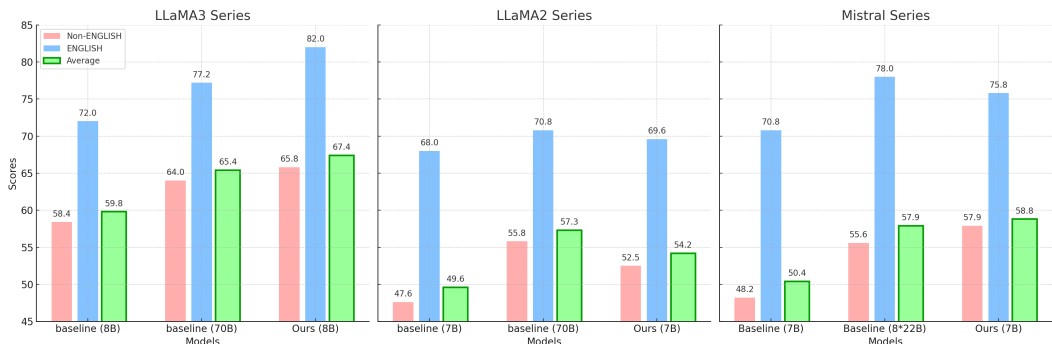

Figure 8: Evaluation results on the MGSM dataset. The score in the image represents the average of the LLM's scores across the 10 languages.

## 4.2 MODEL EVALUATION

In this section, we fine-tune several LLMs on the HighMath-350k dataset and compare their performance against baseline models. We also extend our approach to solving mathematical problems using python code, demonstrating the generalizability and effectiveness of our method.

### 4.2.1 MATHEMATICAL REASONING

**Baseline Models** For the MGSM dataset, we selected state-of-the-art fine-tuned baselines from (Zhu et al., 2024b;a). Additionally, for the MSVAMP evaluation, we included baselines derived from fine-tuning LLaMA2 series models. Where direct comparisons were not feasible, we established an additional baseline using a dataset of 350k samples evenly distributed across 10 languages to ensure consistency.

**Main Results** The results, depicted in Figures 8 and 9, demonstrate that models fine-tuned on the HighMath-350k dataset achieved an average accuracy improvement of 12% on the MGSM evaluation dataset compared to the baselines. Furthermore, these models outperformed larger parameter models from the LLaMA3 and Mistral series by 8%, despite using only 70% of the training data volume required by baseline models.Furthermore, compared to methods that uniformly translate English data into multiple languages, our approach saves over 70% in translation costs.

### 4.2.2 SOLVING MATHEMATICAL REASONING TASKS USING PYTHON CODE

Using the HighCode-350k dataset, which focuses on solving mathematical problems through Python code, our models consistently outperformed the baselines across multiple base models. This demonstrates that optimized language proportions enhance both natural and code-based reasoning, highlighting the versatility of our fine-tuning approach.

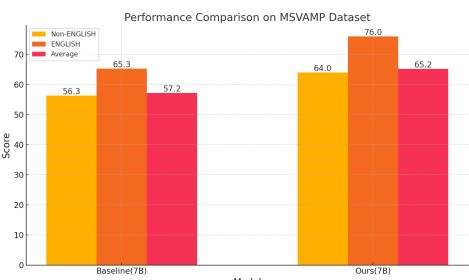 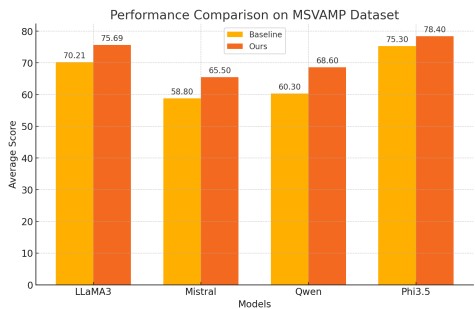

Figure 9: In the left image, the baseline represents (Zhu et al., 2024a), and non-English refers to the average performance across 9 non-English languages. In the right image, the baseline represents the fine-tuning results using an equal amount of data, but with the proportion of different languages in the dataset being balanced. The scores are averaged across 10 languages.

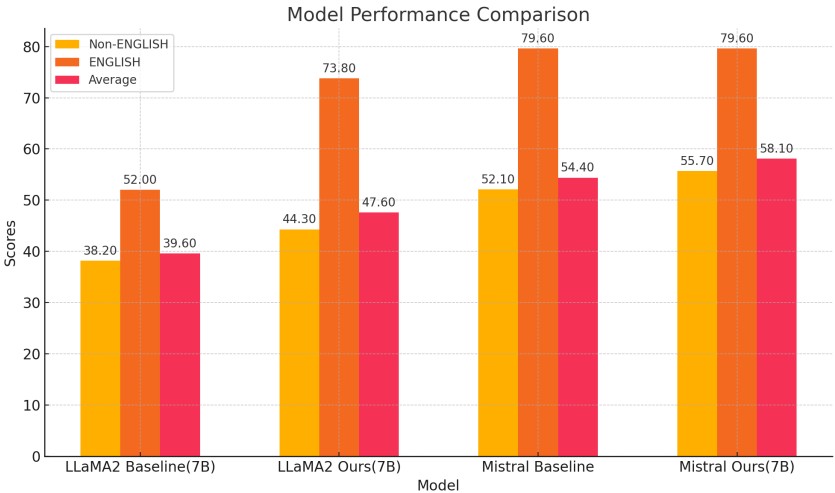

Figure 10: The score in the image represents the average of the LLM's scores across the 10 languages.

## 5 CONCLUSION

For scenarios that require simultaneously enhancing LLMs' reasoning capabilities across multiple languages through fine-tuning methods, we propose an innovative three-phase approach and conduct extensive experiments to investigate the impact of different language data proportions in multilingual training datasets on LLM fine-tuning for multilingual mathematical reasoning tasks. Additionally, we establish a mathematical model to illustrate the relationship between these factors. We also perform large-scale experiments to explore the correlation between fine-tuning effectiveness and the amount of training data in multilingual mathematical reasoning datasets. Using an optimal data scale, we construct the multilingual chain-of-thought reasoning dataset HighMath-350k and the HighCode-350k dataset, designed for solving mathematical problems using Python code in a multilingual context. Leveraging these datasets, we fine-tuned several LLMs to achieve outstanding performance, providing insights for future research.

ACKNOWLEDGEMENTS

The authors would like to express their sincere gratitude to the anonymous reviewers for their insightful review comments. We also extend our sincere thanks to Professor Kehai Chen for his generous support.

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

APPENDIX: GAUSSIAN PROCESS REGRESSION MODEL FITTING AND EVALUATION

INTRODUCTION TO GAUSSIAN PROCESS REGRESSION

## A GAUSSIAN PROCESS REGRESSION: MATHEMATICAL FOUNDATION AND KERNEL SELECTION

### A.1 MATHEMATICAL FOUNDATION OF GAUSSIAN PROCESS REGRESSION

Gaussian Process Regression (GPR) is a non-parametric Bayesian method used for regression problems. It assumes that the observed data can be generated by a Gaussian process, which is a collection of random variables such that any finite subset has a joint Gaussian distribution.

Given a training dataset $\{(X_i, y_i)\}_{i=1}^n$, where $X_i$ represents the input variables (in this case, two-dimensional coordinates) and $y_i$ represents the corresponding observations, we assume:

$$y_i = f(X_i) + \varepsilon_i$$

where $f(X)$ is the latent true function, and $\varepsilon_i \sim \mathcal{N}(0, \sigma_n^2)$ is independent and identically distributed Gaussian noise.

A Gaussian process prior is placed on $f(X)$:

$$f(X) \sim \mathcal{GP}(m(X), k(X, X'))$$

where $m(X)$ is the mean function, typically set to zero, $m(X) = 0$, and $k(X, X')$ is the covariance function (kernel) that measures the similarity between input points.

#### A.1.1 PREDICTIVE DISTRIBUTION

For a new input point $X_*$, the predictive distribution is also a Gaussian distribution with mean and variance given by:

$$\mu_* = k_*^\top (K + \sigma_n^2 I)^{-1} y, \quad \sigma_*^2 = k(X_*, X_*) - k_*^\top (K + \sigma_n^2 I)^{-1} k_*$$

where:

- $y = [y_1, y_2, \ldots, y_n]^\top$ is the vector of observed values.
- $K$ is the kernel matrix, $K_{ij} = k(X_i, X_j)$.
- $k_* = [k(X_*, X_1), k(X_*, X_2), \ldots, k(X_*, X_n)]^\top$ is the covariance vector between the new input point and the training data.
- $I$ is the identity matrix.

### A.2 KERNEL FUNCTION SELECTION AND PARAMETER TUNING

Choosing an appropriate kernel function $k(X, X')$ is crucial for the effectiveness of Gaussian Process Regression. Several kernel functions were considered:

**Radial Basis Function (RBF) Kernel / Gaussian Kernel:**

$$k_{\text{RBF}}(X, X') = \sigma_f^2 \exp\left(-\frac{\|X - X'\|^2}{2l^2}\right)$$

**Reason:** The RBF kernel is smooth and infinitely differentiable, making it suitable for predicting smoothly varying functions. It is one of the most commonly used kernels.

**Matern Kernel (e.g., $\nu = \frac{3}{2}$ or $\nu = \frac{5}{2}$):**

$$k_{\text{Matern}}(X, X') = \sigma_f^2 \frac{2^{1-\nu}}{\Gamma(\nu)} \left( \frac{\sqrt{2\nu} \|X - X'\|}{l} \right)^\nu K_\nu \left( \frac{\sqrt{2\nu} \|X - X'\|}{l} \right)$$

where $K_\nu$ is the modified Bessel function.

**Reason:** The Matern kernel is more flexible than the RBF kernel, allowing control over the smoothness of the function, making it suitable for handling data with varying smoothness.

**Rational Quadratic Kernel:**

$$k_{\text{RQ}}(X, X') = \sigma_f^2 \left( 1 + \frac{\|X - X'\|^2}{2\alpha l^2} \right)^{-\alpha}$$

**Reason:** The Rational Quadratic kernel is a weighted sum of RBF kernels with different length scales, making it suitable for data with multi-scale features.

### A.2.1 PARAMETER TUNING

The hyperparameters of the kernel functions were adjusted as follows:

- **Length Scale $l$:** Values tested: $l = 0.1, 1, 10$.
  **Reason:** The length scale controls the rate of change of the function. Smaller $l$ values allow greater changes over shorter distances, suitable for capturing local features; larger $l$ values result in smoother functions.
- **Signal Variance $\sigma_f^2$:** Values tested: $\sigma_f^2 = 1, 5, 10$.
  **Reason:** The signal variance determines the amplitude of function variation. Adjusting $\sigma_f^2$ helps match the overall variability level of the data.
- **Noise Variance $\sigma_n^2$:** Values tested: $\sigma_n^2 = 0.01, 0.1, 1$.
  **Reason:** The noise variance reflects the level of noise in the observed data. Adjusting $\sigma_n^2$ according to the actual data helps prevent overfitting or underfitting.

**Combining Kernels:** Multiple kernels were also combined, for example, a linear kernel plus an RBF kernel:

$$k(X, X') = k_{\text{Linear}}(X, X') + k_{\text{RBF}}(X, X')$$

**Reason:** Combined kernels can capture different features in the data, such as global trends and local variations.

### A.3 RESULTS EVALUATION METHODS

To evaluate the performance of the GPR models with different kernel functions, several metrics were used:

**Mean Squared Error (MSE):**

$$\text{MSE} = \frac{1}{N} \sum_{i=1}^{N} (y_i - \mu_i)^2$$

**Purpose:** Measures the average squared difference between the predicted and actual values. The smaller the MSE, the better the model performance.

**Root Mean Squared Error (RMSE):**

$$\text{RMSE} = \sqrt{\text{MSE}}$$

**Purpose:** RMSE is in the same units as the original data, making it easier to interpret the magnitude of the error.

**Mean Absolute Error (MAE):**

$$\text{MAE} = \frac{1}{N} \sum_{i=1}^{N} |y_i - \mu_i|$$

**Purpose:** Measures the average absolute difference between the predicted and actual values, less sensitive to outliers than MSE.

**R-squared (Coefficient of Determination):**

$$R^2 = 1 - \frac{\sum_{i=1}^{N}(y_i - \mu_i)^2}{\sum_{i=1}^{N}(y_i - \bar{y})^2}$$

**Purpose:** Evaluates the proportion of variance in the dependent variable that is predictable from the independent variables. The closer $R^2$ is to 1, the better the model fits the data.

## A.4 EXPERIMENTAL RESULTS AND ANALYSIS

The results for different kernel functions are summarized in Table 4. These results indicate the performance of each kernel function in terms of MSE, RMSE, MAE, and $R^2$ score.

Table 4: Performance of Different Kernels

| Kernel | MSE | RMSE | MAE | $R^2$ |
|---|---|---|---|---|
| RBF | 0.6929 | 0.8324 | 0.6692 | 0.8764 |
| Matern ($\nu = 1.5$) | 0.3211 | 0.5666 | 0.4659 | 0.9427 |
| Matern ($\nu = 2.5$) | 0.4152 | 0.6443 | 0.5265 | 0.9259 |
| Rational Quadratic | **0.3199** | **0.5656** | **0.4639** | **0.9429** |

**Discussion:**

- The **RBF kernel** showed relatively poor performance with the highest MSE and RMSE, indicating it might not be flexible enough to handle the non-smooth data in this case.

- The **Matern kernel** ($\nu = 1.5$) performed very well with a low MSE of 0.3211 and a high $R^2$ of 0.9427, suggesting a good balance between model complexity and generalization ability.

- The **Matern kernel** ($\nu = 2.5$) also performed well, but slightly worse than $\nu = 1.5$, indicating its smoothness might not fully match the data characteristics.

- The **Rational Quadratic kernel** achieved the best performance with the lowest MSE (0.3199), RMSE (0.5656), and MAE (0.4639), indicating it is very well-suited to the multi-scale characteristics of the data.

## A.5 CONCLUSION

Based on the experimental results, the **Rational Quadratic kernel** is selected as the optimal kernel function for the Gaussian Process Regression model. It shows the best fitting performance in terms of MSE, RMSE, and MAE metrics. The **Matern kernel** ($\nu = 1.5$) is also a recommended choice when a slightly smoother function is needed.

