# OpenReview forum: "Enhancing Multilingual Reasoning in LLMs: Insights from Cross-Linguistic Correlations and Optimal Data Proportions"
_ICLR.cc/2025/Conference — ICLR 2025 Conference Withdrawn Submission_

### Official Review · Reviewer_wdjh · 2024-10-29

**Soundness:** 3
**Presentation:** 3
**Contribution:** 3
**Rating:** 8
**Confidence:** 4

**Summary:**

LLM gains reasoning ability across multiple languages after finetuning. The typical approach is to use the same amount of data entries in each language during finetuning. This paper points out that it is more efficient and effective when the proportion of each language is different. Thru a 3-phase systematic approach, authors found the optimal balance in terms of the proportions and achieved the SOTA performance in math reasoning and python code reasoning. It effectively reduces the data volume and translation cost needed during finetuning for multilingual reasoning.

**Strengths:**

The paper is well written and nicely presented with comprehensive tables and figures.

The topic discussed is original and significant, especially during the finetuning stage; that is equal amount of data entries in different languages may not result in the optimal performance.

Authors made it clear that via experiments; by strategically distributing the amount of data across multiple languages, the multilingual reasoning ability can be acquired at much cost.

The experiment covers 25 languages and most performant open source LLMs.

Authors created two multilingual datasets, namely HighMath-350k and HighCode-350k.

The idea, "given a fixed amount of data volume, how to achieve the optimal performance by including the right amount of data for each aspect", can be borrowed at different stages of the training process, which will lower the cost of training as well as the GPU hours needed.

**Weaknesses:**

There are some minor grammar issues, such as no space after a comma or a period. For example, line 221 "selection,in ..." and line 231 "system.In ..."

Another minor layout issue is that the last two figures split the conclusion onto two pages.

Figure 3 and figure 7 might need some description briefly explaining the purpose for improved readability.

Figure 3 depicts three scores (COMET, BLEURT, BLEU), but there is no elaboration in the context about them.

It is unclear why the authors only adopted finetuning from non-English to English translation pairs and omitted bi-directional pairs in phase 1.

Finetuning configuration is not properly documented.

The translation of dataset is solely based on the model DeepSeek-Chat-v2-236B. Even though authors adopted some methodology to ensure the quality of translation, like preserving formulas and using Arabic numerals, the validation of the translated content can be improved. For example, a subset sampled from the translation can be further validated by native speakers for coherency and consistency.

**Questions:**

1. what was the reason that English to Non-English translation is not explored?

2. (line 208) what are the two examples used in the prompt?

3. (line 215) what's the percentage of discarded data entries?

4. does the pattern / ratio found in this paper single-purposely apply to math reasoning dataset? could it generalize to other multilingual tasks? or should researchers and developers follow the 3 phases and look for a new ratio?

5. (table 1) why is only llama3 investigated in phase 3?

6. (figure 3) the reasoning performance dropped after finetuning (like Swahili), and the dropped amount is close to the improvements. how should we understand this phenomenon?

7. can you explain the reason why model llama2 was not shown in Table1 but used in phase 3 in the experiment?

---

> ### Author Response · Authors · 2024-11-14
> **Responses to Reviews and Manuscript Revisions**
>
> Thank you for your insightful comments. Your feedback is valuable for improving our paper quality. We have revised the paper according to your suggestions. Here are our responses:
>
> ### Figure 3 shows three scores (COMET, BLEURT, BLEU) without detailed explanation in the text.
>
> Thank you for your suggestion. We have added a caption to Figure 3 and included descriptions of these three metrics in the appendix.
>
> ### Fine-tuning configuration is not properly documented.
>
> Thank you for your suggestion. We have briefly added the fine-tuning configuration in the new version. Detailed configurations will be provided in our public code repository.
>
> ### Improving validation of translated content
>
> We agree with your point. While LLMs have made rapid progress in translation capabilities, having native speakers review a subset would indeed enhance translation quality and validation, making the experiments more rigorous and convincing. However, considering cost constraints and the current strong translation performance of LLMs, we ultimately did not adopt this approach. We appreciate this excellent suggestion and your professional insights.
>
> ### Why wasn't English to Non-English translation explored?
>
> Yes, we agree with your point. Exploring both English to non-English and non-English to English translation in Phase 1 would indeed strengthen the preliminary experiments. Due to computational resource constraints (as it requires fine-tuning multiple language pairs across four models with substantial data), we didn't pursue this approach and acknowledge this limitation in our methodology. We appreciate your attention to detail and for pointing out potential experimental setup improvements.
>
> ### What are the two examples used in the prompt?
>
> The two examples are carefully selected high-quality samples, which will be detailed in our public code repository.
>
> ### What's the percentage of discarded data entries?
>
> Based on our reasoning parameters, the discarded data accounts for 4.03% of total data. This typically relates to parameter settings, which will be detailed in the code.
>
> ### Could it generalize to other multilingual tasks?
>
> Our research scope is limited to mathematical reasoning. Generalization to other domains requires further investigation. However, we believe this would be a valuable research direction to demonstrate our method's generalizability. Thank you for the suggestion.
>
> ### Why was only llama3 investigated in phase 3?
>
> Phase 3 required fine-tuning models with large amounts of data, consuming substantial computational resources, especially considering the 70B parameter model. Additionally, previous experiments showed consistency across various LLMs. For these reasons, we selected the representative llama3 series for experiments while acknowledging this limitation. Your suggestion would indeed make the paper more rigorous, and we appreciate your feedback.
>
> ### How should we understand the dropped reasoning performance after fine-tuning (like Swahili)?
>
> This is an important question. We believe different evaluation metrics focus on different aspects, so some fluctuation across metrics is normal. As you noted, Swahili showed significant decline in COMET and BLEURT metrics but remained relatively stable in BLEU, possibly suggesting that mathematical reasoning data might reduce LLM's performance on Swahili. However, we cannot draw definitive conclusions without further experiments, as this might also simply be a normal fluctuation.
>
> ### Can you explain why llama2 wasn't shown in Table 1 but was used in phase 3?
>
> We did not use llama2 in Phase 3 experiments (Figure 7). We introduced llama2 when comparing our trained models with existing baselines (Figure 8). Since existing baselines were fine-tuned results based on llama2, for fair comparison, we also fine-tuned llama2 and compared it with the baseline. We apologize for any confusion in the paper and will clarify this in the new version.
>
> ## Conclusion
>
> We greatly appreciate your insightful and professional comments, which have significantly helped improve our work. Your recognition greatly encourages us, and we hope our responses and revisions enhance your evaluation of the paper. We welcome further discussion. Thank you!

---

> ### Comment · Reviewer_wdjh · 2024-11-28
>
> Thank you for addressing my all questions. With the changes you mentioned, I believe the paper is now in great shape.

---

> > ### Author Response · Authors · 2024-11-28
> > **Thank you!!!**
> >
> > You can’t imagine how much of a pleasant surprise this was for us! We really appreciate the time you took and your professional advice. Your recognition of our work means a lot and is a huge encouragement for us. We’d be happy to continue the discussion with you. Wishing you all the best, and hoping your paper gets recognized at ICLR. Thank you!

---

### Official Review · Reviewer_gXeb · 2024-10-30

**Soundness:** 3
**Presentation:** 2
**Contribution:** 3
**Rating:** 5
**Confidence:** 3

**Summary:**

This paper presents a systematic study on the impact of language data proportions in multilingual reasoning datasets on the fine-tuning performance of LLMs. The authors claim to have identified optimal language distributions and data volumes for fine-tuning, leading to state-of-the-art performance in multilingual mathematical reasoning and solving mathematical problems using Python code. The study also aims to reduce data volume requirements and translation costs compared to existing methods.

**Strengths:**

The paper addresses a significant gap in the literature regarding the optimal balance of language proportions in multilingual reasoning datasets for LLMs.

The research is methodologically sound, with a clear three-phase approach and over 600 groups of experiments providing a robust basis for the findings.

**Weaknesses:**

The paper outlines a three-phase methodology to minimize the search space to a manageable size. A crucial step involves categorizing languages based on their alignment with English. It raises questions about the general applicability of the findings. If a LLM is trained primarily based on non-English datasets, does it still work?

As stated in the abstract, the paper asserts that it achieves state-of-the-art performance in both multilingual mathematical reasoning and in solving mathematical problems using Python code. However, the experimental section lacks empirical comparisons with existing benchmarks that utilize Python code.

Lastly, the presentation requires further enhancement. The font size of the figures is too small, and it would be beneficial to include descriptions for figures, such as Figure 3 and Figure 7.

**Questions:**

In line 252, Figure 3 depicts the variations in translation performance. Which LLM do the results in the figure pertain to? Is the correlation consistent across different LLMs?

In lines 256 to 259, Figure 4 assesses the model's performance on the MGSM dataset after fine-tuning with various non-English to English translation pairs. However, does the author take into account that fine-tuning with WMT datasets might lead to a decline in mathematical reasoning performance?

In lines 368 to 370, when extending 10 languages to 25 languages, how to get the ratio of en:de:ru:fr:others = 24:8:8:8:12?

---

> ### Author Response · Authors · 2024-11-14
> **Responses to Reviews and Manuscript Revisions**
>
> Thank you for your insightful comments, which have greatly inspired us and helped improve our paper's readability and quality. We appreciate your time and professional advice. We have revised the paper according to your suggestions. Here are our responses:
>
> ### Would the method still work if an LLM is trained primarily on non-English datasets?
>
> Currently, most LLMs are trained primarily on English datasets, so we believe our proposed method still maintains strong generalizability. Although we did not explore this question in the paper, we acknowledge it is a profound and valuable question that would pose higher requirements for the method's universality. We believe investigating this question could reveal more interesting findings and advance the field. Thank you for raising this point, which provides great inspiration for our future work.
>
> ### Regarding Python code benchmarks
>
> For the Python code reasoning experiments, we did compare with existing SOTA baselines (see Figure 10), but we must acknowledge that the baseline selection was not clearly stated in the figure caption. We have revised the unclear description that caused confusion. Thank you for pointing this out.
>
> ### Regarding Figure 3
>
> We apologize for the confusion. While we mentioned the LLMs used in Table 1, we failed to clearly annotate this in the subsequent text. We have made corresponding revisions. Additionally, we have added the raw data and correlation analysis to the appendix. We hope this addresses your concern.
>
> ### Fine-tuning with WMT datasets might lead to decreased mathematical reasoning performance.
>
> This is a valuable question. Since we only fine-tuned the LLM using translation datasets without other measures, the decline in mathematical abilities was expected and normal. As the purpose of this experimental phase was to compare the effects of different languages, the decrease in translation ability does not affect our analysis at this stage. We have added relevant explanations to the paper based on your comment. Thank you for this suggestion.
>
> ### How was the ratio of en:de:ru:fr:others = 24:8:8:8:12 determined?
>
> This is an insightful question. Based on our previous experiments, we found that the proportions of German, French, and Russian should be increased, so we simply increased the ratios of these three languages when expanding to 25 languages. This is a qualitative approach, and we acknowledge its suboptimality. However, overall, this choice still proved effective, demonstrating the validity of our method. We apologize for not explaining this clearly in the paper and have made corresponding revisions.
>
> ## Conclusion
>
> In conclusion, we greatly appreciate your specific corrections to our paper, and your suggestions provide valuable inspiration for our future work. We apologize for any confusion caused by our previous presentation and have revised both the content and layout according to your suggestions to improve the paper's quality and readability. Your evaluation is very important to us, and we sincerely hope our improvements meet your satisfaction. Your approval would be a great encouragement for our efforts. We welcome further discussion and are happy to address any remaining concerns. Thank you!

---

> > ### Comment · Reviewer_gXeb · 2024-11-26
> >
> > Thank you for your feedback. That addresses my problem. I will raise my score to 5.

---

> > > ### Author Response · Authors · 2024-11-26
> > > **Thank you!**
> > >
> > > Thank you very much for your recognition of this paper and your open attitude. Your feedback has been a great encouragement to us. We are also glad to know that our responses have been helpful to you. Your initial comments were highly professional and inspiring, and we have made significant improvements to the paper based on your suggestions.
> > >
> > > Once again, we sincerely appreciate your acknowledgment of our work, and if you have any other concerns, please let us know. We would be happy to discuss further with you. We sincerely wish you all the best and hope that all your papers are successfully accepted. Thank you!

---

### Official Review · Reviewer_o1pT · 2024-11-01

**Soundness:** 4
**Presentation:** 4
**Contribution:** 4
**Rating:** 8
**Confidence:** 4

**Summary:**

The paper shows a way to enhance multilingual reasoning in LLMs by optimizing the proportion of language data in multilingual reasoning datasets. It addresses challenges in fine-tuning LLMs for low-resource languages, which often lack sufficient training data, by investigating the impact of language proportions on model performance. The paper shows optimal language ratios and volume of data for fine-tuning. The paper also present 'HighMath-350k' and 'HighCode-350k' datasets for multilingual mathematical reasoning and Python-based problem-solving.

**Strengths:**

The paper shows a unique way to improve multilingual reasoning in LLMs by optimizing the dataset language proportions and that is highly relevant for AI applications. This work contributes a systematic approach to identifying optimal language data ratios. Also, the paper shows extensive methodology and use of Gaussian Process Regression for language group optimization. The outcome/resultant datasets from this work (HighMath-350k and HighCode-350k) are specifically for multilingual reasoning tasks and that will be helpful for future research in this field.

**Weaknesses:**

There are 25 languages in the phase3 of this study. But in phase 1 and 2, they used a smaller subset. How does this difference may affect the generalizability of findings to low-resource languages is not covered. Also, the datasets generated in this study are based on Non-English to English translation task. However, the study does not cover how to safeguard against translation inaccuracies in low-resource languages.

**Questions:**

1. How do you plan to do quality control for translations in languages where direct verification might be challenging?
I am asking this question because some low-resource languages may not generate accurate translations in English or vice-versa. It would be interesting to know the author's view on this.

---

> ### Author Response · Authors · 2024-11-14
> **Responses to Reviews and Manuscript Revisions**
>
> Thank you for your highly professional review. Your expertise in this field is evident, as you have precisely identified one of the key challenges we encountered during our experiments. We indeed observed that LLMs struggle to provide reasonable translations for some very rare languages (like Sami), likely due to insufficient training data in these languages. Here is our detailed response:
>
> ### How do we control translation quality?
>
> For practical considerations, the 25 languages selected in our study are not extremely low-resource languages, and LLMs can generally provide reasonable translations for them. Beyond the filtering methods mentioned in the paper, we implemented additional safeguards:
>
> Taking Swahili (sw) as an example, we:
> 1. Selected 30 English sentences and had them translated to Swahili using our chosen DeepSeek model
> 2. Used GPT-4 and Claude 3.5-Sonnet to back-translate the Swahili content to English
> 3. Manually evaluated the accuracy of the final English translations to verify whether LLMs could reasonably handle translation for this lower-resource language
>
> While we acknowledge this may not be a perfect solution, we believe this approach provides meaningful quality control. Actually, We have identified the development of more robust methods for translation verification as an important direction for our future research.
>
> Although this method helps evaluate translation quality, improving LLM translation capabilities for very rare languages (like Sami) remains an important research challenge. This might be addressed through fine-tuning approaches, but falls outside the scope of the current study.
>
> ### How does the variation in language selection across phases affect the generalizability of findings to low-resource languages?
>
> We appreciate you raising this important point. We acknowledge that Figure 7 did not explicitly show the performance improvements for each low-resource language individually. We have addressed this by adding detailed performance improvements for all low-resource languages in the appendix of the revised paper, which we hope provides the clarity you sought.
>
> ### Conclusion
>
> In conclusion, we sincerely thank you for your insightful review. We have revised the paper following your suggestions and hope our responses address your concerns satisfactorily. We welcome further discussion. Thank you!

---

> > ### Comment · Reviewer_o1pT · 2024-11-26
> >
> > Thanks for sharing your insights around my question. I think i am going to keep my score intact since it is already high enough.

---

> > > ### Author Response · Authors · 2024-11-27
> > > **Thank you!**
> > >
> > > Thank you so much for your time and insightful feedback. It’s truly an honor to exchange ideas with you. Your support for this paper means a lot to us and is a great source of encouragement. We would be delighted to continue our discussions with you. Wishing you all the best, and hoping all your papers are successfully accepted by ICLR!

---

### Official Review · Reviewer_rXzQ · 2024-11-04

**Soundness:** 2
**Presentation:** 2
**Contribution:** 2
**Rating:** 5
**Confidence:** 4

**Summary:**

This paper investigates how varying language proportions in multilingual reasoning datasets impact the fine-tuning performance of large language models (LLMs). The authors aim to demonstrate that carefully balancing language data can enhance LLM performance, achieving state-of-the-art results in multilingual mathematical reasoning and problem-solving while reducing data and translation costs.

**Strengths:**

There is significant interest in optimizing the pre-training or fine-tuning mixtures for multilingual LLMs, and this work aims to provide valuable insights into this area.

**Weaknesses:**

W0: The overall writing of the paper lacks coherence, making it difficult to follow the experimental phases. The fragmented presentation of the experiment phases obscures essential aspects of the study, such as training characteristics, the mixture used for fine-tuning, the evaluation tasks, metrics, and analysis. Additionally, several figures, such as Figure 3, lack complete captions, contributing to the confusion. There is also repetition in section headers (e.g., "Phase 2" in Lines 269 and 320), which affects readability and structure.


There are a lot of points and claims that lock in motivation and support:
- W1: Why is language alignment only studied within the context of mathematical reasoning? The rationale for limiting the focus to this particular setting is unclear.
- W2: What criteria were used to select the 10 or 25 languages? How are "high resource" languages defined in this context? Is the selection based on a specific pretraining dataset, and is there a reference for this?
- W3: The grouping of languages appears arbitrary. What was the basis for forming these specific language groups?
- W4: In Line 189, why did the authors choose only high-resource languages for language alignment when, as stated in Line 35, the problem primarily impacts low-resource languages? This choice seems to contradict the motivation expressed in the introduction.
- W5: In Line 45, the paper claims, "The key is to efficiently leverage a small amount of low-resource language data to broadly enhance the multilingual reasoning capabilities of LLMs." Is there any literature or existing research that supports this assertion?

W6: It appears that the evaluation was conducted solely on the MGSM benchmark. This raises concerns about overfitting the optimal mixture to this specific benchmark. How can the proposed method for creating an "optimal mixture" be generalized to other benchmarks?

**Questions:**

Q1: Many references are missing from the claims presented in section 1/ introduction

Q2: Figure 2: Is this data format known for the literature about its performance? Where is the reference?

Q3: Line 076: The motivation for the language extension from 10 to 25 is not clearly explained. What does it mean that the authors extended the language coverage to 25 languages?

Q4: It might be better to provide the full language names in the tables instead of the language codes since, depending on the implementation and the taxonomy used, the codes might be different.

Q5: Please provide captions on all the figures.

---

> ### Author Response · Authors · 2024-11-14
> **Responses to Reviews and Manuscript Revisions**
>
> Thank you for your insightful comments, which have greatly helped us improve our work. We have polished our paper according to your suggestions. Here are our responses to your concerns:
>
> 1. Regarding W1 - Why is language alignment only studied within mathematical reasoning?
> Our primary objective was to enhance LLMs' mathematical reasoning capabilities across multiple languages. We first investigated the relationship between multilingual alignment and mathematical reasoning. After achieving success in mathematical reasoning, we applied this approach (using specific multilingual training data proportions) to other reasoning scenarios and found similarly positive results. We acknowledge that our original explanation may have caused confusion and have revised it in the new version.
>
> 2. Regarding W2 - Criteria for selecting 10 or 25 languages:
> a) Selection of 10 languages: These languages are commonly used in related works ( [1][2][3] and so on. ) to evaluate LLMs' multilingual mathematical reasoning capabilities. This selection enables both representation and comparison with existing research.
>
> b) Extension to 25 languages: To further validate our method's effectiveness, we expanded beyond the conventional 10 languages to include 25 languages, covering both commonly used and low-resource languages. The selection referenced historical WMT datasets for better representation. We have added this explanation to the paper for clarity.
>
> 3. Regarding W3 - Basis for language grouping:
> Our grouping followed these principles:
> - English forms one group due to its dominance in training data and strongest reasoning capabilities
> - German, French, and Russian form another group based on their demonstrated positive correlation with English alignment and mathematical reasoning abilities
> - The remaining languages form the third group
> We have clarified this explanation in the revised paper.
>
> 4. Regarding W4:
> Our approach enhances reasoning capabilities across all languages (including low-resource ones) by increasing training data for high-resource languages (de, fr, ru) in fine-tuning datasets. We chose high-resource languages for alignment study because of their abundant, readily available data, which helps investigate how to generalize English reasoning capabilities to other languages. We have revised our explanation to make this clearer.
>
> 5. Regarding W5:
> Given the scarcity and high translation costs of low-resource language data, coupled with LLMs' relatively poor reasoning performance in low-resource contexts, efficient utilization of existing low-resource data is crucial. We have added relevant citations and clarified this explanation in the paper.
>
> 6. Regarding W6 - Generalization to other benchmarks:
> Your concern is valid. We addressed this by:
> - Using both MGSM and MSVAMP datasets for evaluation (Figure 9)
> - Extending our method to Python code reasoning
> - These are the primary benchmarks in the field, also used by related works ([1][2][3] and so on ).
> - We will open-source our code, datasets, and models for community feedback
>
> Regarding Q1 - Missing references:
> Thank you for pointing this out.  We have added the relevant citations in Section 1 to make our arguments more rigorous and convincing.
>
> Regarding Q2 - Data format reference:
> Thank you for pointing this out. The data format has been validated in previous research, and we have added the missing reference to Figure 2.
>
> Regarding Q3 - Language extension motivation:
> We have clarified that the extension from 10 to 25 languages was aimed at further validating our method's effectiveness. While the initial 10 languages were representative, expanding to 25 languages allowed us to test our approach more comprehensively across both high-resource and low-resource languages.
>
> Regarding Q4 - Language names:
> We appreciate this helpful suggestion and have added a comprehensive table in the appendix that maps language codes to their full names.
>
> Regarding Q5 - Figure captions:
> Thank you for pointing this out. We have updated all figures with complete and descriptive captions as suggested.
>
> ### Conclusion
> In conclusion, we sincerely appreciate your valuable feedback, which has helped us significantly improve our paper. We have implemented your suggestions and hope our revisions and responses provide a clearer understanding of our work. Your recognition is important to us, and we earnestly hope to receive a more favorable evaluation based on these modifications. We welcome further discussion and are happy to address any remaining concerns. Thank you!
>
>
>
> [1] The Power of Question Alignment in Multilingual Reasoning: Broadened Scope and Deepened Insights
>
> [2] Question Translation Training for Better Multilingual Reasoning
>
> [3] MindMerger: Efficiently Boosting LLM Reasoning in non-English Languages

---

> > ### Comment · Reviewer_rXzQ · 2024-11-25
> >
> > Thank you for your response and the additional comments. I have decided to keep my original scores.

---

> ### Author Response · Authors · 2024-11-25
>
> We greatly appreciate your time and professional feedback. Following your advice, we have greatly improved the quality of this paper. We regret that our initial response did not improve your evaluation of this paper, but we respect your decision. If there is anything we can do to further address your concerns, please let us know. Your advice has truly helped us a lot. It is our honor to discuss with you. Thank you!

---

### Public Comment · ~Léo_Durand1 · 2024-11-15
**Comments from a Researcher with Similar Experimental Experience**

I have read over 10 papers from ICLR 2025 that are related to my research area, and I must say this is the best one in my view. I find the three-phase approach proposed in this paper highly innovative.

In fact, I previously had similar ideas about exploring the optimal proportions of different languages in reasoning datasets. However, since it required directly exploring the optimal ratios among more than 10 variables, the enormous search space made the experiments infeasible. Consequently, I only obtained experimental results for a few proportions using the ChatGLM model. Although this differs from the models used by the authors, my experimental results largely align with the patterns they identified. This suggests that the authors' experimental conclusions have strong generalization capability.

Moreover, I think the most innovative aspect of the paper is first using translation tasks to explore the alignment relationship between different languages and English to divide the 10 languages into 3 groups, and then determining the optimal relationship between these 3 variables. I wish I had thought of this approach at the time!

I also have a few suggestions: I would like to know whether this method is effective for tasks other than mathematical reasoning. If it is, this could be a very promising research direction. Additionally, I hope the authors could include more raw experimental data in the appendix.

---

> ### Author Response · Authors · 2024-11-17
>
> Thank you for your high praise of this paper! We are delighted to learn that our experimental conclusions can corroborate each other, which undoubtedly enhances the credibility of this work. As for whether the methods mentioned in this paper can be extended to fields beyond mathematical reasoning, this indeed goes beyond the scope of our research. However, I believe Dmitry's comments might provide some valuable insights.
>
> Additionally, due to the large volume of data, we will provide more raw experimental data in the code repository to facilitate the reproducibility of this research. Thank you!

---

### Public Comment · ~Dmitry_Olegovich_Smirnov1 · 2024-11-16
**I think I might have an answer**

Since both reviewers are curious about whether this method can be applied to fields outside of mathematics, I actually ran some experiments in the LLM instruction-following area before, so I might have some ideas to share.

I tested the multilingual instruction-following ability using the LLaMA2-7B model. What I found was that fine-tuning on data for non-English languages (like Russian, French, Chinese, etc.) did improve the model’s ability to follow instructions in all the test languages. But I didn’t do large-scale experiments to dig deeper into this because, like Leo, I wasn’t sure how to reduce the search space. After reading this paper, I think the author’s approach and conclusions could be extended to the instruction-following domain. It still needs more experimentation, but I see a lot of potential in this direction.

Overall, I think the author’s methods are really innovative and the results are impressive. The writing and formatting in this version are good. I’d give this paper a 9, maybe even a 10. Because this article can corroborate other experimental results, I think that is important.

---

> ### Author Response · Authors · 2024-11-17
>
> Thank you so much for generously sharing your experimental results and for your high recognition of our work! Your sharing has greatly helped us and given us more confidence in the methods we proposed. Thank you for your comment!

---

### Public Comment · ~Yijiu_Chen1 · 2024-11-16

Thank you for the excellent work! I noticed that the authors utilized a large-scale multilingual mathematical reasoning dataset in the paper. To my knowledge, the field of multilingual mathematical reasoning has consistently lacked large amounts of high-quality data, as translating such data requires significant financial and labor resources. The scarcity of datasets has hindered many research efforts.

I would like to know if the authors plan to make the dataset used in the paper publicly available. If the dataset could be released, I believe it would be a significant contribution to the community and greatly enhance the impact of the paper.

I look forward to hearing back from the authors.

---

> ### Author Response · Authors · 2024-11-17
> **Will the dataset be open-sourced?**
>
> Thank you for your comment! Yes, we will make the large-scale dataset used in the paper publicly available. We hope this will enable other researchers to reproduce our work and further advance the field. You will have access to the dataset in the near future, and I will notify you when it becomes available.

---

> ### Public Comment · ~Yijiu_Chen1 · 2024-11-17
>
> Thank you in advance to the authors for their generous sharing! I believe this paper is undoubtedly excellent, and I wish you the best of luck!

---

### Note · Authors · 2025-04-08

**Comment:**

### Attention

Due to unresolved disputes regarding the affiliated institutions and authors' names, we regret to inform you that this paper has been withdrawn. Please note that the retraction is due to the authorship controversy and does not reflect on the correctness of its content.

**Withdrawal Confirmation:**

I have read and agree with the venue's withdrawal policy on behalf of myself and my co-authors.

---

### Meta-Review · Area_Chair_YG2k · 2024-12-23

**Metareview:**

This paper examines the optimal balance of multilingual reasoning datasets in the post training mixture of LLMs.  The authors provide extensive experiments to show the various tradeoffs.  Reviewers overall like the paper (with two reviewers strongly supporting acceptance).  I believe that this paper will be quite useful for the community that is investigating multilingual fine tuning of models hence I support acceptance.  On the weakness front, I think abiding by what Reviewer rXzQ is suggesting in terms of writing, will improve the paper.

**Additional Comments On Reviewer Discussion:**

There has been significant discussion between the authors and reviewers and in some cases the reviews and corresponding responses helped improve the paper.

---

### Decision · Program_Chairs · 2025-01-22

Accept (Poster)